# Assessing the efficacy of albendazole against hookworm in Vietnam using quantitative PCR and sodium nitrate flotation

**Clare E. F. Dyer** [1] *, **Naomi E. Clarke**[1], **Dinh Ng Nguyen**[2], **H. M. P. Dilrukshi Herath**[3], **Sze Fui Hii**[3], **Russell Pickford**[4], **Rebecca J. Traub**[3], **Susana Vaz Nery**[1]

**1** Kirby Institute, University of New South Wales, Sydney, Australia, **2** Faculty of Animal Sciences and Veterinary Medicine, Tay Nguyen University, Đắk Lắk, Vietnam, **3** Faculty of Veterinary and Agricultural Sciences, University of Melbourne, Victoria, Australia, **4** Bioanalytical Mass Spectrometry Facility, University of New South Wales Analytical Centre, Sydney, Australia

* cdyer@kirby.unsw.edu.au

**Data Availability Statement:** All relevant data are within the manuscript and its Supporting Information files.

## Abstract

Preventive chemotherapy (PC), consisting of the regular distribution of anthelmintics to populations or groups of populations at risk, is the primary tool used to control soil-transmitted helminth (STH) infections. This strategy, whilst cost-effective, raises the concern of potential emergence of drug resistance. The efficacy of anthelmintics against STH infections is measured using cure rate (CR) and egg reduction rate (ERR), using microscopy-based techniques such as the Kato-Katz thick smear. However, Kato-Katz has low sensitivity, especially for low-intensity infections, and requires fresh samples that need to be processed quickly. Realtime quantitative PCR (qPCR), which is more sensitive, is emerging as a "gold standard" for STH diagnostics given its higher sensitivity (important in low prevalence settings) and ability to differentiate hookworm species, while sodium nitrate flotation (SNF) may provide a low-cost more sensitive and practical alternative to Kato-Katz in the field. In this study, we examined the efficacy of a locally manufactured brand of albendazole 400 mg ("Alzental") against hookworm in Đắk Lắk province, Vietnam, using both qPCR and SNF. For qPCR, formulae to convert qPCR cycle threshold (Ct) values into eggs per gram of faeces (EPG) were utilised to determine efficacy calculations, and these values directly compared with efficacy values generated using SNF. Factors associated with CR and ERR were examined, and Alzental tablet quality was assessed by comparing with an Australian TGA-approved equivalent "Eskazole" tablet. We observed a CR and ERR of 64.9% and 87.5% respectively using qPCR, and 68.4% and 67.6% respectively using SNF. The tablet composition of Alzental was comparable to Eskazole in terms of active albendazole drug concentration with no evidence of impurities. This study demonstrates that the efficacy of Alzental against hookworm is within the range of previously reported studies for albendazole 400 mg. The study also demonstrates the value of qPCR and SNF as alternatives to standard Kato-Katz methodology for assessment of anthelmintic efficacy.

**Funding:** The study was funded as part of an NHMRC grant awarded to S.V.N for the Community Deworming against soil-transmitted helminths (CoDe-STH) trial (grant number APP1139561). The funders had no role in study design, data collection and data analysis, decision to publish, or preparation of the manuscript.

**Competing interests:** The authors have declared that no competing interests exist.

## Author summary

Regular administration of deworming drugs to whole at-risk populations is the recommended strategy to control soil-transmitted helminth (STH) infections in humans. Repeated rounds of deworming drug administration could lead to emerging drug resistance.

In this study, we examined the efficacy of the deworming drug albendazole 400 mg ("Alzental") against hookworm in Đắk Lắk province, Vietnam. Efficacy is measured by calculating cure rates and egg reduction rates after Alzental treatment. We also compared two different diagnostic methods for detecting and quantifying STH infections–a microscopy approach with sodium nitrate flotation, and a molecular approach with quantitative PCR.

We show the efficacy of Alzental is within the range previously reported for albendazole and we found no evidence of emerging drug resistance. SNF and qPCR may provide more convenient, and more sensitive alternatives to the current standard diagnostic tool (the Kato Katz thick smear).

## Background

Soil-transmitted helminths (STHs) are parasitic worms that affect around 900 million people worldwide [1]. They include *Necator americanus*, *Ancylostoma duodenale* and *Ancylostoma ceylanicum* (hookworms), *Ascaris lumbricoides* and *Strongyloides stercoralis* (roundworms), and *Trichuris trichiura* (whipworm). Chronic STH infections contribute to malnutrition and iron-deficiency anaemia, impair physical and cognitive development, and can lead to poor school performance in children, reduced work productivity in adults, and adverse pregnancy outcomes [2,3]. Soil-transmitted helminthiasis is classified by the World Health Organization (WHO) as one of the twenty neglected tropical diseases (NTDs) [4].

The WHO advocates regular school-based delivery of the benzimidazole anthelmintic drugs albendazole or mebendazole once or twice annually to school-age children as the main strategy to control STH-associated morbidity [5,6]. Vietnam's Ministry of Health has implemented a school-based deworming program since 2000, and in Đắk Lắk province in the Central Highlands of Vietnam, large-scale deworming of primary school-age children with albendazole or mebendazole has taken place since 2007 [7]. The deworming program uses albendazole and mebendazole donations from GlaxoSmithKline (GSK) and Johnson & Johnson, but in-country production of other brands also occurs. At least 16 different brands of albendazole and 10 different brands of mebendazole are manufactured by different pharmaceutical companies with different pricing and availability countrywide, and sold over the counter [8]. A survey conducted by the World Health Organization (WHO) to identify possible quality control problems in NTD drugs found that 41 out of 72 samples collected in the three South-East Asia countries studied, including Vietnam, failed a dissolution test, and in one case an albendazole sample also failed a content of active ingredient and uniformity of dosage units test [9,10]. This raises concerns about drug quality and efficacy given the number of different brands available in Vietnam.

There are limited albendazole efficacy data from Vietnam, with two studies using the GSK brand "Zentel" [11,12], and one using a locally-manufactured brand "Mekozetel" (Mekophar Chemical Pharmaceuticals) [13]. Further studies are needed to assess the efficacy and chemical composition of other brands of albendazole used in Vietnam, in the context of an ongoing long-term deworming program.

Development of anthelmintic resistance is a concern in countries with long-term preventive chemotherapy programs given that drug resistance is now reported in every livestock host, and for every anthelmintic drug class [14,15]. The WHO recommends regular assessment of drug efficacy in STH control programs to monitor for potential emergence of drug resistance [16]. The majority of efficacy studies have used Kato-Katz, or other microscopy methodology such as McMaster and FLOTAC techniques, to determine cure rate (CR) and egg reduction rate (ERR). However, these techniques are known to have reduced sensitivity where egg counts are low [17,18]. Quantitative PCR (qPCR) is emerging as the "gold standard" for STH diagnostics, especially in settings where infection intensity is lower after multiple rounds of preventative chemotherapy, because it is more sensitive and able to detect very low-level infections, and also allows differentiation between hookworm species [19]. However, only a few studies have used qPCR to determine efficacy of anthelmintic drugs, and only one has calculated infection intensity in eggs per gram of faeces [20–22]. As qPCR becomes more popular as a diagnostic tool for STH, more studies are needed that compare its diagnostic performance with microscopy, especially because WHO classification of heavy-, moderate-, and light-intensity infections are based on microscopy parameters [23]. Furthermore, whilst Kato-Katz is the most commonly used diagnostic tool in resource-limited settings, the logistical challenges of collecting and processing fresh samples within a short time-frame due to hookworm eggs embryonating and becoming prone to rupture in glycerol, further drives the need to validate alternative microscopy tools that are more sensitive, yet still practical and affordable. Sodium nitrate flotation (SNF), a microscopy technique that has been widely used in veterinary parasitology, may be such an alternative, as samples can be preserved in a fixative upon collection and later examined in a central laboratory, and it has been shown to be more sensitive than Kato Katz [24–27].

In this study, we use both SNF and qPCR diagnostic techniques to assess the efficacy of a locally manufactured brand of albendazole 400mg, Alzentel (Shin Poong Pharmaceuticals), in Vietnam and compare the sensitivity and diagnostic agreement of these two approaches. We also look at factors associated with CR and ERR using both techniques and compare the chemical composition of Alzental with the Australian Therapeutic Goods Administration (TGA)-approved equivalent Eskazole to investigate tablet quality.

## Methods

### Study design

A parasitological survey was conducted across four hamlets in Đắk Lắk province, Vietnam, in April–May 2020. Four villages were randomly selected from a total of 64 villages already involved in the Community Deworming against STH (CoDe-STH) trial underway in the region [7]. One hamlet was randomly selected from each of the four villages, together with a back-up, for inclusion in the study.

### Sample size calculation

To detect normal versus reduced efficacy based on faecal egg count (FEC) reduction, a sample size of at least 200 individuals (irrespective of infection status) is recommended by the WHO, with at least 50 positive individuals [23,28]. Based on a 2018 prevalence survey conducted in Đắk Lắk province in school-age children, we assumed a hookworm prevalence of at least 10% in each of six age groups (1–5 years, 6–11 years, 12–17 years, 18–29 years, 30–49 years, 50 + years). A non-sample return rate of 25% across both timepoints was assumed, therefore we aimed to enrol 40 individuals per age group (a total of 240) per hamlet in the study and expected to collect 32 stools per age group in each hamlet, to a total of 128 samples per age group, and 768 samples overall. Based on an average of 4 individuals per household,

60 households per hamlet were selected initially, with 20 additional households selected in case more were required to achieve the target sample size for every age group. Based on the baseline estimated prevalence this would provide more than the 50 positive samples recommended by the WHO for efficacy assessment.

Given the expected very low prevalence of *A. lumbricoides*, *T. trichiura*, and *S. stercoralis*, this study was not powered to determine efficacy against these parasites.

### Field procedures

Households were randomly selected from a list of all households in the community, obtained from the hamlet leader. All community members were eligible to participate, excluding women in their first trimester of pregnancy, children aged <1 year, and individuals in poor health [29]. Each selected household was visited, and all eligible household members invited to participate. Written informed consent was obtained for all household members over 18 years of age, and parental informed written consent was obtained for household members under 18 years. Once 40 individuals were invited from an age band, no further individuals from that age band were invited in that hamlet.

The following day, stool samples were collected and subsequently assessed for infection status and infection intensity. Individuals who provided a stool sample were given a single dose of Alzental (400mg albendazole). Children aged 12–23 months old received 200mg. All doses were taken under direct observation of the research team and recorded in a treatment register. For ethical reasons, all eligible individuals living in a participating household but who did not provide a stool sample were also offered albendazole treatment. At 14 days post-treatment, all individuals who provided a stool sample and who took Alzental under direct observation were asked to provide a second stool sample.

### Stool sample preparation and analysis

Two aliquots of each stool sample were preserved by adding 3g of stool to 3 mL 5% potassium dichromate and immediately chilling on ice. One aliquot was analysed by SNF in duplicate as described previously [24,25] using sodium nitrate of specific gravity 1.20. The other aliquot was sent to the University of Melbourne, Australia, where quantitative multiplex real-time PCR analysis was used to detect and quantify infections as described previously [25,26,30,31]. Of note is that in SNF eggs are floated from approximately 1 g faeces, whereas in PCR DNA is extracted from 0.2 g faeces.

Approximately 10% of the qPCR-positive samples were screened under the microscope to determine the embryonation status of STH eggs as a result of storage and transportation. Samples with the highest infection intensity (the lowest Ct values) were purposefully selected. Samples were found to be embryonated, and therefore a previously-derived Ct-to-EPG conversion formula for embryonated *N. americanus* eggs was used (EPG = $10^{(Ct-32.657)/-3.878}$) [32]. The equation was derived as follows: a serial dilution of known quantities of *N. americanus* eggs were seeded into parasite-free faeces and incubated at 28°C for 5 days to allow for embryonation of eggs to occur [32]. A standard curve of Ct versus EPG values was then generated, following the same procedures published previously for unembryonated eggs [30]. Samples collected after albendazole treatment were only analysed if the corresponding sample collected before treatment tested positive for any species of STH, consistent with the WHO protocol for efficacy calculations [23].

### Statistical analysis

Data collected at each of the two study time points were analysed to determine the prevalence and intensity in eggs per gram (EPG) of faeces of each STH species by qPCR and SNF.

qPCR was considered the "gold standard" for the purpose of sensitivity calculations. Specificity was assumed to be 100%. Diagnostic agreement between qPCR and SNF was assessed using Cohen's Kappa coefficient ($\kappa$).

For infection intensity calculations, Ct to EPG conversion equations were only available for *N. americanus* and not *A. ceylanicum* or *A. duodenale* hookworm species. Since *N. americanus* is the dominant hookworm species in the region, for the purposes of comparing qPCR with SNF results, we assumed that all hookworm infections detected by SNF were caused by *N. americanus*.

Efficacy was calculated in terms of both cure rate and egg reduction rate. Only individuals who were initially hookworm positive were included in these calculations [23].

Cure rate (CR) was calculated using the following formula [23]:

$$\frac{No.\ individuals\ positive\ before\ treatment\ and\ negative\ after\ treatment}{No.\ individuals\ positive\ before\ treatment} \times 100$$

Egg reduction rate (ERR) was calculated using the following formula [23]:

$$100 \times \left( 1 - \frac{Mean\ egg\ counts\ after\ treatment}{Mean\ egg\ counts\ before\ treatment} \right)$$

ERR was calculated using both arithmetic and geometric mean egg counts. 95% binomial exact confidence intervals were calculated for CR and prevalence, and confidence intervals for ERR were calculated using a bootstrap re-sampling method with 10,000 replicates.

Mixed-effects logistic regression models were used to examine factors associated with cure and with infection intensity reduction for *N. americanus*, by both SNF and qPCR. Univariable analyses were carried out for age, gender, and infection intensity before albendazole treatment, with variables retained for inclusion in the multivariable model if $p < 0.25$. Models were adjusted for clustering at the hamlet level.

All statistical analyses were carried out using Stata version 16.1 (College Station, TX, USA).

## Chemical analysis of albendazole tablet

Three Alzental and three Eskazole tablets were selected at random and powdered using a pestle and mortar. 1 mg of each powdered tablet brand was then dissolved in 1mL formic acid. A 500 µg/mL albendazole standard was also prepared in formic acid. Each was diluted 40,000 times in methanol 0.1% formic acid for liquid chromatography–tandem mass spectrometry (LC-MS/MS) [33,34]. Dilution was required to ensure signals were within the range of the instrument.

The expected µg/mL concentration for each tablet brand was determined by taking the mass of albendazole and the mass of the whole tablet to calculate the percentage of the tablet that is albendazole, and therefore what the expected µg/mL would be. The drug recovery (actual versus theoretical µg/mL) was calculated for each tablet and the albendazole standard. Output from the LC-MS/MS was also examined for any impurities in the tablets. Eight replicate analyses were conducted for each tablet to ascertain accurate instrument error.

## Ethical considerations

Ethical approval for this study was granted from the Human Research Ethics Committees at the University of New South Wales, Australia (HC180740) and Tay Nguyen University, Vietnam (1804/QĐ-ĐHTN-TCCB).

A small reward in the form of 20,000 VND (~AU$1) phone credit was given to each participant after the second stool sample was collected.

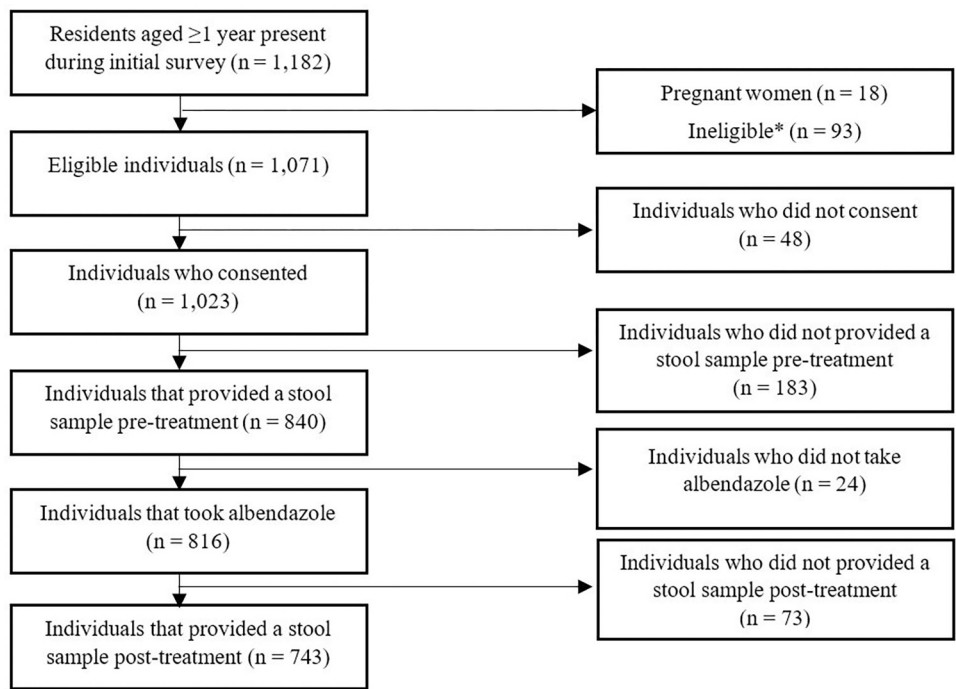

**Fig 1. Efficacy study design.** *Ineligible due to poor health as determined by the research team.

## Results

### Study population

From the four hamlets selected to participate in the study, 1,182 individuals were present in households randomly selected to participate, of which 1,071 were eligible for the study and 1,023 (95.5%) consented to participate (**Fig 1**). Of these, 743 individuals provided a stool sample at both timepoints (before and after albendazole treatment) and took albendazole after providing the initial sample.

The mean age of the 743 participants was 25.8 years, ranging from 1 to 93 years, with 47.1% male and 52.9% female.

### Prevalence of STH before albendazole treatment

As shown in Table 1, the most prevalent STH identified by qPCR and SNF was hookworm (prevalence 40.2% (95% CI 36.8–43.8) and 33.7% (95% CI 15.6–58.3) respectively), with prevalence of hookworm higher by qPCR than SNF ($p = 0.010$). *N. americanus* was the most prevalent species of hookworm diagnosed by qPCR (prevalence 39.8% (95% CI 19.2–64.8)). *A. ceylanicum* was only found in one hamlet (prevalence <5%), with no evidence of *A. duodenale* in any of the four hamlets surveyed. Hookworm prevalence (all species) increased with increasing age (qPCR: OR = 1.87, p<0.001; SNF: OR = 1.73, p<0.001; **S1 Fig**). The overall prevalence in adults aged 50+ across the four hamlets was 62.0% (95% CI 18.5–92.1), while prevalence in children aged 1–5 years was 6.2% (95% CI 1.4–23.2). Proportions and trends were comparable between the two diagnostic methods (**S1 Fig**).

The arithmetic mean infection intensity for *N. americanus* by qPCR was 1,204.3 EPG, ranging from <1 to 36,596 EPG (**Table 1**). By SNF, the arithmetic mean was 133.6 EPG,

**Table 1. Prevalence and infection intensity of different species of STH before albendazole treatment by qPCR and SNF.**

|  | qPCR | SNF |
|---|---|---|
| **Prevalence, % (95% CI)** | **N = 743** | **N = 743** |
| **Any STH** | 42.1 (19.2–69.0) | 34.5 (16.1–59.2) |
| *A. lumbricoides* | 0.3 (0.0–1.6) | 0.5 (0.1–3.1) |
| *T. trichiura* | 2.3 (0.1–31.8) | 0.5 (0.0–11.6) |
| *S. stercoralis* | 4.2 (1.4–11.4) | N/A |
| Hookworm spp. | 40.2 (36.8–43.8) | 33.7 (15.6–58.3) |
| *N. americanus* | 39.8 (19.2–64.8) | N/A |
| *A. ceylanicum* | 1.2 (0.0–23.3) | N/A |
| *A. duodenale* | 0 | N/A |
|  |  |  |
| **Co-infections** |  |  |
| **Any coinfections** (%, (*95% CI*)) | 5.1 (1.2–19.4) | 0.3 (0.0–5.8) |
|  |  |  |
| *Hookworm spp/A. lumbricoides* | 0.1 (0.0–3.0) | 0 |
| *Hookworm spp/T. trichiura* | 1.2 (0.1–16.1) | 0.3 (0.0–5.8) |
| *Hookworm spp/S. stercoralis* | 3.4 (1.3–8.7) | N/A |
| *N. americanus/A. lumbricoides* | 0.1 (0.0–3.0) | N/A |
| *N. americanus/T. trichiura* | 1.1 (0.1–13.8) | N/A |
| *N. americanus /S. stercoralis* | 3.1 (1.2–7.5) | N/A |
| *N. americanus /A. ceylanicum* | 0.8 (0.0–16.6) | N/A |
|  |  |  |
| **>2 species (%, *95% CI*)** | 0.5 (0.0–11.6) | 0 |
|  |  |  |
| **Infection intensity category** [35,36] |  |  |
| Hookworm spp. |  |  |
| Heavy | N/A | 0 (0%, -) |
| Moderate | N/A | 0 (0%, -) |
| Light | N/A | 250 (33.8%, 30.5–37.3) |
| *N. americanus* |  |  |
| Heavy | 23 (3.1%, 2.1–4.6) | N/A |
| Moderate | 23 (3.1%, 2.1–4.6) | N/A |
| Light | 250 (33.7%, 30.3–37.1) | N/A |
| **Infection intensity, EPG (range)** | **N = 296** | **N = 250** |
| Hookworm spp. | N/A | 133.6 (2–1,185) |
| *N. americanus* | 1,204.3 (<1–36,596) | N/A |

ranging from 2 to 1,185 EPG. Using WHO thresholds for determining infection intensity, prevalence of light intensity infections was 33.7% and 33.8% by qPCR and SNF respectively, and prevalence of high and moderate intensity infections was 6.2% and 0% by qPCR and SNF respectively (**Table 1**). Of infected individuals, by qPCR 84.5% of infections were light intensity, with 15.4% high and moderate intensity. By SNF, 100% hookworm infections were light intensity.

Prevalence of *A. lumbricoides* and *T. trichiura* was very low. No statistically significant difference in prevalence between diagnostic methods was seen for *A. lumbricoides* (0.3% by qPCR versus 0.5% by SNF) or *T. trichiura* (2.3% by qPCR versus 0.5% by SNF). *S. stercoralis* had an overall prevalence of 4.2% by qPCR.

Coinfections were detected in 40 individuals. By qPCR, the most common was coinfection with *N. americanus* and *S. stercoralis* (25 individuals), followed by *N. americanus* and *T. trichiura* (10 individuals), *N. americanus* and *A. ceylanicum* (6 individuals), and one individual was coinfected with *N. americanus* and *A. lumbricoides*. Four individuals were infected with three or more species of STH. All individuals testing positive for hookworm species by qPCR had *N. americanus* (either alone or as a coinfection).

Prevalence and coinfection prevalence were calculated as a percentage (and adjusted 95% CI) of the total paired samples available (n). Infection intensity data only shown for hookworm

**Table 2. Diagnostic agreement between qPCR and SNF, and sensitivity of SNF versus qPCR for diagnosing hookworm infection.**

| | | qPCR | | Mean EPG by qPCR if qPCR+ N = 64 | Mean EPG by SNF if qPCR- N = 16 | Agreement n (%) | Kappa statistic* | | Sensitivity** |
| | | Positive | Negative | | | | κ | P value | SNF |
|---|---|---|---|---|---|---|---|---|---|
| SNF | Positive | 235 | 16 | 1,532 | 40.3 | 663 (89.2%) | 0.77 | <0.001 | 78.6% (235/299) |
| | Negative | 64 | 428 | 16.5 | - | | | | |

*Kappa agreement classification: <0.20 = poor; 0.21–0.40 = fair; 0.41–0.60 = moderate; 0.61–0.80 = good; 0.81–1.00 = very good

**Sensitivity defined as true positives/total positives x 100

due to very low prevalence of other STH species. Ct to EPG conversion equations only available for *N. americanus* species of hookworm for qPCR data. N/A: Not applicable.

### Diagnostic agreement between qPCR and SNF

The sensitivity of SNF compared with qPCR, and the diagnostic agreement between the two methods, were calculated for hookworm infection for the 743 participants who provided samples before treatment, with both diagnostic results available. Sensitivity was 78.6% (235/299) for SNF compared with qPCR. The diagnostic agreement between SNF and qPCR was good (κ = 0.77, p < 0.001) (**Table 2**).

Of note is that 16 out of 250 samples that tested positive for hookworm by SNF, before albendazole treatment, tested negative for hookworm by qPCR. These samples had a mean EPG by SNF of 40.3 EPG (range 2–273 EPG) (**Table 2**). Conversely, there were 64 out of 296 samples that tested positive for hookworm by qPCR that were missed by SNF. The mean infection intensity of these 64 samples by qPCR was 16.5 EPG (range <1–466 EPG), which was significantly lower than samples testing positive by both qPCR and SNF at 1,532 EPG (range <1–36,596 EPG, p<0.001) (**Table 2**).

### Cure rate and factors associated with cure for hookworm

Table 3 shows that the cure rate for hookworm by qPCR was 64.9% (adjusted 95% CI 48.9–78.0%). Cure rate by SNF was comparable (*p* = 0.383) at 68.4% (adjusted 95% CI 60.0–75.7).

Age, gender, and infection intensity before albendazole treatment were not associated with being cured of hookworm (**S1 Table**).

**Table 3. Cure rates for hookworm by qPCR and SNF overall, and by sex and age group.**

| | qPCR | | | SNF | | |
| | N | Number cured | Cure rate, % *(95% CI)* | N | Number cured | Cure rate, % *(95% CI)* |
|---|---|---|---|---|---|---|
| **Overall** | 296 | 192 | **64.9** (48.9–78.0) | 250 | 171 | **68.4** (60.0–75.7) |
| **Sex** | | | | | | |
| Male | 148 | 93 | **62.8** (46.8–76.5) | 127 | 82 | **64.6** (55.2–72.9) |
| Female | 148 | 99 | **66.9** (49.4–80.7) | 123 | 89 | **72.4** (57.1–83.7) |
| **Age group** | | | | | | |
| 1–5 | 7 | 5 | **71.4** (4.6–99.2) | 5 | 5 | **100** |
| 6–11 | 23 | 16 | **70.0** (2.6–99.5) | 18 | 12 | **66.7** (43.2–84.0) |
| 12–17 | 40 | 24 | **60.0** (39.2–77.7) | 35 | 24 | **68.6** (14.9–96.5) |
| 18–29 | 67 | 51 | **76.1** (54.6–89.4) | 56 | 43 | **76.8** (51.3–91.2) |
| 30–49 | 79 | 49 | **62.0** (25.7–88.5) | 70 | 45 | **64.3** (37.3–84.5) |
| ≥50 | 80 | 47 | **58.8** (41.5–74.1) | 66 | 42 | **63.6** (47.9–76.9) |

Adjusted for clustering at the hamlet level (adjusted 95% CI shown); N: Number of people infected before treatment.

**Table 4. Infection intensity values before and after albendazole treatment, and ERR as measured by qPCR for *N. americanus* and SNF for hookworm.**

| | | Pre-treatment mean infection intensity (EPG) Mean (95% CI) | Post-treatment mean infection intensity (EPG) Mean (95% CI) | ERR (%) (95% CI) |
|---|---|---|---|---|
| **Arithmetic means** | **qPCR** | 1,204.3 (842.1–1,566.5) | 150.6 (92.0–209.2) | **87.5** (81.4–93.6) |
| | **SNF** | 133.6 (108.9–158.3) | 43.4 (18.7–68.2) | **67.3** (51.3–83.8) |
| **Geometric means** | **qPCR** | 128.2 (92.7–177.4) | 63.1 (38.2–104.3) | **50.8** (25.9–75.7) |
| | **SNF** | 54.3 (45.4–65.0) | 35.0 (24.3–50.4) | **35.8** (12.5–59.1) |

### Egg reduction rate (ERR) of *N. americanus* and associated factors

The ERR by qPCR for the 296 people infected with *N. americanus* before albendazole treatment was 87.5% (95% CI 81.4–93.6) from an arithmetic mean infection intensity of 1,204.3 EPG (95% CI 842.1–1,566.5) to 150.6 EPG (95% CI 92.0–209.2). By SNF the ERR for the 250 people infected with hookworm species before albendazole treatment was lower at 67.3% (95% CI 51.3–83.8), though the pre-treatment mean infection intensity was much lower than by qPCR at only 133.6 EPG (95% CI 108.9–158.3) EPG. Using geometric means for the calculations resulted in a lower ERR by both qPCR and SNF at 50.8% and 35.8% respectively (**Table 4**).

S2 Fig shows the distribution of individual ERRs for *N. americanus* measured by qPCR and SNF. Most infections were cured or had an ERR of at least 80% according to both diagnostic methods. By qPCR, 75.4% of *N. americanus* infections had an ERR of at least 80%, with 64.9% of infections cured by qPCR. Of the 104 people still infected with *N. americanus* after albendazole treatment, 36 had a higher intensity after albendazole treatment compared to before. The results were comparable by SNF, where 77.6% of *N. americanus* infections had an ERR of at least 80%, with 68.4% of infections cured; of 79 people still infected with *N. americanus* after albendazole treatment by SNF, 25 had a higher infection intensity after treatment.

Age was not associated with ERR for *N. americanus* by either qPCR or SNF (**S2 Table**). Gender was not associated with *N. americanus* ERR by SNF, but females had a very slightly higher ERR than males by qPCR.

### Albendazole tablet composition

No obvious differences between Eskazole and Alzental were observed during the tablet analysis (**S3 Table**). The expected mass of albendazole in each tablet was calculated based on the mass of the whole tablet. The percentage recovery of albendazole from Alzental (65.1%) was not significantly different from the recovery of albendazole from Eskazole (67.2%) (p = 0.160). There was no evidence of tablet impurities in either tablet brand (**S3 and S4 Figs**).

### Discussion

This study adds to the growing evidence base of how different techniques–in this case qPCR and SNF–compare in their ability to diagnose STH infections. We also show that a single dose of Alzental albendazole 400 mg is efficacious in the treatment of hookworm in children and adults living in Đắk Lắk province, Vietnam, and this locally manufactured brand was of similar quality to an Australian-approved equivalent "Eskazole".

This study is novel in its use of qPCR to determine anthelmintic drug efficacy by converting Ct into EPG values that may be directly compared with microscopy results and applied to WHO infection intensity classification. The few existing efficacy studies that have employed qPCR diagnostics have used relative fluorescent units or DNA concentration as a proxy for EPG values [21,22]; to date only one other study based in Cambodia has taken a similar

approach to ours in using Ct to EPG conversion equations to examine efficacy of albendazole by qPCR versus SNF [20].

The CRs we estimate by qPCR and SNF diagnostic tools are within the range of CRs estimated in other studies. A systematic review and meta-analysis of thirty-eight studies estimated a cure rate of 79.5% (95% CI 71.5–85.6) for albendazole against hookworm species [17]. We speculate that the cure rates we report (64.9% by qPCR, and 68.4% by SNF) may be lower than the meta-analysis findings because of the diagnostic tools we used, rather than an indication of emerging drug resistance or issues with drug quality [25,37]. qPCR has a higher diagnostic sensitivity than microscopy-based approaches, and can detect even very light hookworm infections, which would be missed using microcopy [25]. SNF also has a more sensitive diagnostic ability compared with Kato-Katz, especially in detecting hookworm infections [24,25]. Kato-Katz is the most commonly used technique in efficacy studies, but cure rates may be over-estimated since individuals with residual low-intensity infections may be misclassified as cured [25,38].

The observed higher infection intensity observed in qPCR versus SNF results pre-treatment is notable (1,204 vs 133.6 EPG respectively, but this difference is consistently reported when comparing microscopy-based EPG values with Ct to EPG conversions for qPCR [25,26,30]. Additional work is needed to further validate conversion of qPCR obtained Ct values into EPG values.

The WHO recommends use of ERR as the appropriate indicator of drug efficacy [12,23,38]. The ERR we report for Alzental against hookworm by qPCR and SNF differed between the two diagnostic techniques, at 87.5% by qPCR and 67.3% by SNF when using arithmetic means as the measure of central tendency, consistent with current WHO recommendations [23]. A systematic review and meta-analysis of thirty-eight studies reported an ERR of 89.6% (95% CI 81.9–97.3) for albendazole against hookworm species; results were adjusted for whether arithmetic or geometric mean was used [17]. The ERR we calculated by qPCR is within this range, while the ERR calculated by SNF is somewhat lower. A previous study also found that arithmetic mean ERR was higher by qPCR compared with SNF at 72.5% and 40.8% respectively [20]. The lower ERR by SNF may be due to low-level infections being missed by SNF before treatment but detected by qPCR. After treatment, the "missed" sample by SNF would be excluded from the ERR calculation (as the participant was deemed "not infected") but included in the qPCR ERR calculation. A low-level infection before treatment, that was cured after treatment, would show as a 100% reduction in infection intensity for that participant, resulting in a higher ERR by qPCR than SNF. However, when using geometric mean egg counts, the ERR was much lower for both methods, at 50.8% by qPCR and 35.8% by SNF. There is an ongoing debate as to the advantages and disadvantages of arithmetic versus geometric means for such calculations, however, the WHO currently recommends use of the arithmetic mean for such calculations [23].

Use of qPCR to diagnose STH infections is emerging as the "gold standard", due to its proven ability to diagnose low-intensity STH infections, and to differentiate between hookworm species [25]. This is an important consideration in countries aiming for elimination of STH transmission, where multiple rounds of preventative chemotherapy mean most infections are low-intensity [19]. Indeed, we found that the mean infection intensity of samples positive by qPCR, but negative by SNF, was significantly lower than that of samples positive by both methods, indicating the low intensity infections were missed by SNF. However, in a resource limited setting, SNF can be a favorable alternative to Kato-Katz given its superior sensitivity in detecting STH, particularly hookworm eggs compared with Kato-Katz, and that diagnostics are performed on either preserved or fresh fecal samples, simplifying fieldwork procedures [24,39]. In addition, preserving samples slows degradation of hookworm eggs [40]. Two previous studies demonstrated a "moderate" and "fair" Kappa statistic agreement for hookworm diagnosis for SNF compared with qPCR, [26,37], and here

we have demonstrated a "good" agreement in a similar setting, supporting the use of SNF as an superior field-based alternative to Kato-Katz.

Of note are the 16 samples that were positive for hookworm species by SNF and negative by qPCR before albendazole treatment, which we believe are "false positives" by SNF and not qPCR "false negatives". We cannot rule out that in a very low intensity infection, the fecal aliquot used for qPCR analysis could indeed be devoid of STH eggs whilst the aliquot used for SNF could contain eggs, especially since the volume of fecal aliquot used for SNF is five times that for qPCR. However, we believe the presence of other helminth species, namely *Trichostrongylus* and *Meloidogyne* eggs, which have a morphology similar to hookworm, albeit larger in size, and are easily confused with hookworm eggs [41–43], is the main contributor to false positives by microscopy. *Meloidogyne* eggs have been documented in human stool samples in this province of Vietnam [44,45], and *Trichostrongylus* infection in humans has been reported in neighboring Thailand and Laos [46–48]. Further work is underway to identify these species more definitively using qPCR techniques.

This study has several limitations in addition to the potential species misclassification described. Due to the delays in the sample shipment arriving from Vietnam, samples for qPCR embryonated during transportation–despite preserving samples in potassium dichromate and placing them on ice in the field, eggs may still embryonate if not consistently kept refrigerated until DNA extraction [21]. Whilst Ct to EPG conversion equations were specifically developed for embryonated samples and used here, it is not known whether all samples in the batch were fully/equally embryonated. Due to operational constraints, we were not able to conduct Kato Katz microscopy on the samples, which would have provided useful additional diagnostic comparison. Additionally, whilst the 14-day follow-up period used here is consistent with WHO guidelines, we cannot rule out the possibility of residual STH genetic material being amplified by qPCR resulting in a false positive, which would mean we are underestimating the CR. Finally, we were limited in the available variables to test in our risk factor analysis. Only very limited sociodemographic, and no water, sanitation, and hygiene information was collected due to the design of the study, which primarily focused on efficacy and diagnostic comparison.

In conclusion, using qPCR and SNF, we have demonstrated that efficacy of the anthelmintic drug Alzental manufactured in Vietnam is within the previously reported range. In the future this will provide a useful reference point for albendazole efficacy as Vietnam continues its regular preventative chemotherapy program, and the need for ongoing monitoring of emerging resistance becomes ever more critical. This study provides further real-world evidence as to the potential for SNF as a cost-effective, more practical alternative to Kato Katz in the field, providing a useful estimate of community prevalence, even when infection intensity is low. However, qPCR still holds significant advantages over microscopic tools for STH diagnoses, such as its higher sensitivity and the ability to accurately diagnose different species of STH.

## Supporting information

**S1 Fig. Hookworm prevalence (and adjusted 95% CI) by age group diagnosed by qPCR and SNF.**
(TIF)

**S2 Fig. Distribution of individual *N. americanus* infection intensity reduction rates determined by qPCR and SNF.**
(TIF)

**S3 Fig. LC-MS chromatograms.** (a) Base Peak Chromatogram–Standard, (b) Base Peak Chromatogram–Reference Tablet, (c) Base Peak Chromatogram–Test Tablet, (d) Extracted

Ion Chromatogram—m/z 266.1 (albendazole) vs time (standard sample).
(TIF)

**S4 Fig. High resolution and accurate mass spectrum of albendazole.**
(TIF)

**S1 Table. Results of univariate logistic regressions for infection cure for hookworm by qPCR and SNF.**
(DOCX)

**S2 Table. Results of univariable and multivariate linear regression for *N. americanus* ERR (%).**
(DOCX)

**S3 Table. Summary of sample analysis conducted on Alzental and Eskazole tablets, and an albendazole standard.**
(DOCX)

**S1 File. Personal communication (R. Traub).**
(PDF)

## Acknowledgments

We would like to thank Patsy Zendejas for her work on developing the Ct-to-EPG conversion equations used in this study. We also thank and acknowledge Professor Alex Loukas and Paul Giacomin from James Cook University for providing the *Necator* eggs used in developing these equations.

## Author Contributions

**Conceptualization:** Naomi E. Clarke, Susana Vaz Nery.

**Data curation:** Clare E. F. Dyer.

**Formal analysis:** Clare E. F. Dyer.

**Funding acquisition:** Susana Vaz Nery.

**Investigation:** Dinh Ng Nguyen, H. M. P. Dilrukshi Herath, Sze Fui Hii, Russell Pickford.

**Methodology:** Naomi E. Clarke, Russell Pickford, Rebecca J. Traub, Susana Vaz Nery.

**Project administration:** Naomi E. Clarke, Dinh Ng Nguyen, Rebecca J. Traub, Susana Vaz Nery.

**Resources:** Russell Pickford, Rebecca J. Traub, Susana Vaz Nery.

**Supervision:** Rebecca J. Traub, Susana Vaz Nery.

**Visualization:** Russell Pickford.

**Writing – original draft:** Clare E. F. Dyer.

**Writing – review & editing:** Naomi E. Clarke, Rebecca J. Traub, Susana Vaz Nery.

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
