## [Decision Letter · Decision Letter 0]

18 Mar 2022

Dear Dr Dyer,

Thank you very much for submitting your manuscript "Assessing the efficacy of albendazole against hookworm in Vietnam using quantitative PCR and sodium nitrate flotation" for consideration at PLOS Neglected Tropical Diseases. As with all papers reviewed by the journal, your manuscript was reviewed by members of the editorial board and by several independent reviewers. In light of the reviews (below this email), we would like to invite the resubmission of a significantly-revised version that takes into account the reviewers' comments. 

Please address the reviewer's questions about methods and data presentation in addition to the minor grammatical points.

We cannot make any decision about publication until we have seen the revised manuscript and your response to the reviewers' comments. Your revised manuscript is also likely to be sent to reviewers for further evaluation.

Sincerely,

Keke C Fairfax, PhD

Deputy Editor

Keke Fairfax

Deputy Editor

Please address the reviewer's questions about methods and data presentation in addition to the minor grammatical points.

Reviewer's Responses to Questions

**Key Review Criteria Required for Acceptance?**

**Methods**

-Are the objectives of the study clearly articulated with a clear testable hypothesis stated?

-Is the study design appropriate to address the stated objectives?

-Is the population clearly described and appropriate for the hypothesis being tested?

-Is the sample size sufficient to ensure adequate power to address the hypothesis being tested?

-Were correct statistical analysis used to support conclusions?

-Are there concerns about ethical or regulatory requirements being met?

Reviewer #1: Further detail is required on the methods used to determine the embryonation - CT - EPG relationships; see below for more detail

Reviewer #2: See below

Reviewer #3: The objectives of the study are clearly described and the study design appears appropriate to address the stated objectives. The population is clearly defined and the size of the study populations is in agreement with respective WHO recommendations. Overall the statistical analyses do support the conclusion and ethical approval was obtained.

**Results**

-Does the analysis presented match the analysis plan?

-Are the results clearly and completely presented?

-Are the figures (Tables, Images) of sufficient quality for clarity?

Reviewer #1: (No Response)

Reviewer #2: See below

Reviewer #3: The results adhere to the study plan. They are clearly and complete presented. The fiqures are of sufficient quality for clarity.

**Conclusions**

-Are the conclusions supported by the data presented?

-Are the limitations of analysis clearly described?

-Do the authors discuss how these data can be helpful to advance our understanding of the topic under study?

-Is public health relevance addressed?

Reviewer #1: more discussion is required in several areas; see below for more detail

Reviewer #2: See below

Reviewer #3: The conclusions are generally supported by the presented data. There are some issues raised in the attached pdf which the authors should address.

**Editorial and Data Presentation Modifications?**

Reviewer #1: (No Response)

Reviewer #2: see below

Reviewer #3: (No Response)

**Summary and General Comments**

Reviewer #1: The paper describes the use of SNF and qPCR to assess the effectiveness of albendazole against hookworm in Vietnam. I think the study is sound, however more data, information and discussion needs to be provided in several areas. 

Major issues

1) Much more information needs to be provided on the development of the formulae for accounting for the effects of embryonation status on the CT to EPG conversion. Some experimental data needs to be provided here, followed by a description of the development of the equations, and how they performed with experimental samples of known EPG and embryonation status. 

Further, more information needs to be given on the embryonation screening process (line 139). For example, how many eggs screened per sample. 

2) the difference in pre-treatment EPGs using the two methods is very striking: 1204 vs 133.6 This needs to be discussed.

3) Further discussion is also required as to why the ERRs differed so much between the two methods. Can the lower sensitivity of SNF (line 309) really explain this ??. Would missing the very low EPG samples really reduce the ERR that much ??

 4) does SNF really provide ‘’an accurate reflection of infection status’ (line 349) when ’64 out of 299 samples that tested positive for hookworm by qPCR ..were missed by SNF” (line 231) ??

Minor issues:

1) Line 19 “albeit effective’ is far too simple a statement to make regarding the PC strategy. Drug treatment may remove a proportion of the worm population in each individual, or cure some individuals, but is this really an effective strategy ? The word ’effective” can’t be used on its own like this without some explanation of what aspects are indeed effective. 

2) Some uncertainty about the number of separate samples examined in the tablet comparison. Line 172-173 states that ‘1mg of each tablet brand’ suggesting only one sample per brand was analyzed. Table S4 heading and footnote suggests that the 3 tablets for each brand were prepared separately, , and then each was examined 8 times by LC-MS. The degree of replications needs to be made clear in the Methods section. 

3) Should the amount of the phone credit (line 186) be stated ???

4) Line 143 should show ref 24 (Zendejas) as well as 30. But see my point above about the inadequacy of just giving these references for the calculation of embryonation – CT – EPG relationships.

Reviewer #2: General: Dyer and co-authors present a study of albendazole treatment efficacy against hookworm, comparing two different diagnostic tools and both cure rates and egg reduction rates. The study is well designed and the manuscript reads very well, with clear focus and appropriate detail and balance between sections. This is an excellent study and manuscript. A number of points are offered for consideration to further strengthen the manuscript.

- The benzimidazole treatment history in the study villages should be summarized (drugs used, periodicity and number of distribution cycles, targeted population segment, treatment coverage etc.)

- The methods mentions a model to identify factors associated with cure but not which factors were considered

- How were albendazole tables selected that were analyzed for the quality control part of the study?

- Mention in the discussion (limitations?) why Kato-Katz was not performed in parallel as reference test? It remains the recommended test for soil-transmitted helminth surveys

- The 14-day follow-up period for drug efficacy assessment is relatively short. The possibility of false-positive PCR results due to residual genetic material should be discussed

- Was any pattern beyond low egg counts observed in the participants with discordant diagnostic results? Were the same individuals discordant pre- and post-treatment? Were PCR negative but parasitologically positive participants cured (with both methods) after treatment or remained positive (if so: in which method)?

- Check the spelling of Trichuris trichiura throughout the manuscript – this reviewer has counted at least 3 variants

Reviewer #3: This manuscript presents a concise study which provides convincing data on the efficacy of an albendazole product produced locally i.e. in Vietnam. Furthermore, by comparatively examining stool samples by qPCR and a coproscopic method new data is obtained on the agreement of these two methods for the detection of hookworm eggs. The scope of the study is overall limited. Although the data on the efficacy of the tested albendazole product are certainly new, these data don't add anything fundamental to current knowledge. However, the data on the comparative use of qPCR and SNF faecal analysis arguably novel (see comments). So overall the paper provides additional data on the field efficacy of albendazole treatments against natural hookworm infections and on the comparative use of microscopic and molecular egg counts used to assess these data.

PLOS authors have the option to publish the peer review history of their article (what does this mean?). If published, this will include your full peer review and any attached files.

Reviewer #1: No

Reviewer #2: Yes: Peter Steinmann

Reviewer #3: No
---

## [Decision Letter · Decision Letter 1]

23 Aug 2022

Dear Dr Dyer,

We are pleased to inform you that your manuscript 'Assessing the efficacy of albendazole against hookworm in Vietnam using quantitative PCR and sodium nitrate flotation' has been provisionally accepted for publication in PLOS Neglected Tropical Diseases.

Best regards,

Keke C Fairfax, PhD

Section Editor

Keke Fairfax

Section Editor

Reviewer's Responses to Questions

**Key Review Criteria Required for Acceptance?**

**Methods**

-Are the objectives of the study clearly articulated with a clear testable hypothesis stated?

-Is the study design appropriate to address the stated objectives?

-Is the population clearly described and appropriate for the hypothesis being tested?

-Is the sample size sufficient to ensure adequate power to address the hypothesis being tested?

-Were correct statistical analysis used to support conclusions?

-Are there concerns about ethical or regulatory requirements being met?

Reviewer #1: see below

Reviewer #2: No comments

**Results**

-Does the analysis presented match the analysis plan?

-Are the results clearly and completely presented?

-Are the figures (Tables, Images) of sufficient quality for clarity?

Reviewer #1: see below

Reviewer #2: No comments

**Conclusions**

-Are the conclusions supported by the data presented?

-Are the limitations of analysis clearly described?

-Do the authors discuss how these data can be helpful to advance our understanding of the topic under study?

-Is public health relevance addressed?

Reviewer #1: see below

Reviewer #2: no comments

**Editorial and Data Presentation Modifications?**

Reviewer #1: (No Response)

Reviewer #2: no comments

**Summary and General Comments**

Reviewer #1: I think that the authors have done a good job in responding to the comments I made on the originally-submitted version of the paper. I also believe that they have responded adequately to the points raised by the other reviewers.

Reviewer #2: The authors have adequately taken into account the comments made by this reviewer

PLOS authors have the option to publish the peer review history of their article (what does this mean?). If published, this will include your full peer review and any attached files.

Reviewer #1: No

Reviewer #2: **Yes: **Peter Steinmann

---

## [Editor Report · Acceptance letter]

25 Oct 2022

Dear Dr Dyer,

We are delighted to inform you that your manuscript, "Assessing the efficacy of albendazole against hookworm in Vietnam using quantitative PCR and sodium nitrate flotation," has been formally accepted for publication in PLOS Neglected Tropical Diseases.

Best regards,

Shaden Kamhawi

co-Editor-in-Chief

Paul Brindley

co-Editor-in-Chief
